# Analysis of different propagation models for the estimation of the topside ionosphere and plasmasphere with an Ensemble Kalman Filter

Tatjana Gerzen[1], David Minkwitz[2], Michael Schmidt[1], Eren Erdogan[1]

[1] Technical University Munich (TUM), Deutsches Geodätisches Forschungsinstitut (DGFI), Arcisstr. 21, Munich, Germany

[2] Airbus Defence and Space, Robert-Koch-Str. 1, Taufkirchen, Germany

*Correspondence to*: Tatjana Gerzen (tatjana.gerzen@tum.de)

**Abstract.**

The accuracy and availability of satellite-based applications like GNSS positioning and remote sensing crucially depend on the knowledge of the ionospheric electron density distribution. The tomography of the ionosphere is one of the major tools to provide link specific ionospheric corrections as well as to study and monitor physical processes in the ionosphere and plasmasphere. In this work, we apply an Ensemble Kalman Filter (EnKF) approach for the 4D electron density reconstruction of the topside ionosphere and plasmasphere with the focus on the investigation of different propagation models and compare them with the iterative reconstruction technique SMART+. The STEC measurements of eleven LEO satellites are assimilated into the reconstructions. We conduct a case study on a global grid with altitudes between 430 and 20200 km, for two periods of the year 2015 covering quiet to perturbed ionospheric conditions. Particularly, the performance of the methods to estimate independent STEC and electron density measurements from the three Swarm satellites is analysed. The results indicate that the methods EnKF with Exponential decay as the propagation model and SMART+ perform best, providing in summary the lowest residuals.

## 1    Introduction

The ionosphere is the charged part of the upper atmosphere extending from about 50 - 1000 km and going over in the plasmasphere. The characteristic property of the ionosphere is that it contains sufficient free electrons to affect the propagation of trans-ionospheric radio signals, as from telecommunication, navigation or remote sensing satellites, by refraction, diffraction and scattering. Therefore, the knowledge of the three-dimensional electron density distribution and its dynamics are of practical importance. Around 50% of the signal delays or range errors of L-band signals used in GNSS originate from altitudes above the ionospheric F2 layer, consisting of topside ionosphere and plasmasphere (cf. Klimenko et al., 2015; Chen and Yao, 2015). So far, especially the topside ionosphere and plasmasphere is not well described.

The choice of the ionospheric correction model has an essential impact on the accuracy of the estimated ionospheric delay and its uncertainty. A widely used approach for ionospheric modelling is the single-layer model, whereby the ionosphere is projected onto a two-dimensional (2D) spherical layer, typically located between 350 and 450 km. However, usually 2D models are not accurate enough to support high accuracy navigation and positioning techniques in real time (cf. e.g. Odijk 2002; Banville 2014). More accurate and precise positioning is achievable by considering the ionosphere as 3D medium. There are several activities in the ionosphere community aiming to describe the mean ionospheric behavior by the development of 3D electron density models based on

long-term historical data. Two widely used models are the International Reference Ionosphere model (IRI, cf. Bilitza et al., 2011) and the NeQuick model (cf. Nava et al., 2008).

Since those models represent a mean behavior, it is essential to update them by the assimilation of actual ionospheric measurements. There is a variety of approaches developed and validated for the ionospheric reconstruction by the combination of actual observations with an empirical or a physical background model. Hernandez-Pajares et al. (1999) present one of the first GNSS-based data-driven tomographic models, which considers the ionosphere as a grid of three-dimensional voxels and the electron density within each voxel is computed as a random walk time series. The voxel-based discretisation of the ionosphere is further used for instance in Heise et al., 2002; Wen et al., 2007; Gerzen and Minkwitz, 2016; Gerzen et al., 2017; Wen et al., 2020. These authors reconstruct the 3D ionosphere by algebraic iterative methods. An alternative is to estimate the electron density as a linear combination of smooth and continuous basis functions, like e.g. spherical harmonics (SPH) (Schaer 1999), B-splines (Schmidt et al., 2008; Zeilhofer, 2008; Zeilhofer et al., 2009; Olivares-Pulido et al., 2019), B-splines and trigonometric B-splines (Schmidt et al. 2015), B-splines and Chapman functions (Liang et al., 2015 and 2016), and empirical orthogonal functions and spherical harmonics (Howe et al., 1998).

Besides the algebraic methods, also techniques taking benefit of information on spatial and temporal covariance information, such as Optimal Interpolation, Kalman Filter, three- and four-dimensional variational techniques and Kriging, are applied to update the modelled electron density distributions (cf. Howe et al., 1998; Angling et al., 2008; Minkwitz et al., 2015 and 2016; Nikoukar et al., 2015; Olivares-Pulido et al., 2019).

Moreover, there are approaches based on physical models, which combine the estimation of the electron density with physical related variables such as neutral winds or the oxygen/nitrogen ratio (cf. Wang, et al. 2004; Scherliess et al., 2009; Lee et al., 2012; Lomidze et al., 2015; Schunk, et al., 2004 and 2016; Elvidge and Angling, 2019).

In general, the majority of data, available for the reconstruction of the ionosphere and plasmasphere, are Slant Total Electron Content (STEC) measurements, i.e. the integral of the electron density along the line of sight between the GNSS satellite and receiver. Often, STEC measurements provide limited vertical information and hence the modelling of the vertical the electron density distribution is hampered (cf. e.g. Dettmering, 2003). The estimation of the topside ionosphere and plasmasphere poses a particular difficulty since direct electron density measurements are rare and since low plasma densities at these high altitudes contribute only marginally to the STEC measurements. Especially, ground-based STEC measurements are dominated by electron densities within and below the characteristic F2 layer peak. Consequently, information about the plasmasphere is difficult to extract from ground-based STEC measurements (cf. e.g. Spencer and Mitchell, 2011). Thus, in the presented work, we concentrate on the modeling of the topside part of the ionosphere and plasmasphere and utilize only the space-based STEC measurements.

In this paper, we introduce an Ensemble Kalman Filter to estimate the topside ionosphere and plasmasphere based on space-based STEC measurements. The propagation of the analysed state vector to the next time step within a Kalman Filter is a key challenge. The majority of the approaches, working with EnKF variants, uses physic-based models for the propagation step (cf. e.g. Elvidge and Angling 2019; Codrescu et al., 2018; Lee et al., 2012). In our work, we investigate the question how the propagation step can be realized, if a physical model is not available or if the usage of a physical model is rejected as computationally time consuming. We discretize the ionosphere and the plasmasphere below the GNSS orbit height by 3D voxels, initialize them with electron densities calculated by the NeQuick model and update them with respect to the data. We present different methods how to perform the propagation step and assess their suitability for the estimation of electron density. For this purpose, a case study

over quiet and perturbed ionospheric conditions in 2015 is conducted, investigating the capability of the estimates
to reproduce assimilated STEC as well as to reconstruct independent STEC and electron density measurements.
We organize the paper as follows: Section 2 describes the EnKF with the different propagation methods and the
generation of the initial ensembles by the NeQuick model. Section 3 outlines the validation scenario with the
applied data sets. Section 4 presents the obtained results. Finally, we conclude our work in Section 5 and provide
an outlook on the next steps.

## 2    Estimation of the topside ionosphere and plasmasphere

### 2.1    Formulation of the underlying inverse problem

The information about the STEC, along the satellite-to-receiver ray path $s$ can be obtained from multi-frequency
GNSS measurements. In detail, STEC is a function of the electron density $Ne$ along the ray path $s$, given by

$$STEC_s = \int Ne(h, \lambda, \varphi)ds,\tag{1}$$

where $Ne(h, \lambda, \varphi)$ is the unknown function describing the electron density values depending on altitude $h$,
geographic longitude $\lambda$ and latitude $\varphi$.
The discretization of the ionosphere by a 3D grid and the assumption of a constant electron density function within
a fixed voxel allows the transformation of Eq. (1) into a linear system of equations

$$STEC_s \approx \sum_{i=1}^{K} Ne_i \cdot h_{si} \Rightarrow y = Hx + r,\tag{2}$$

where $y$ is the $(m \times 1)$ vector of the STEC measurements, $x$ is the vector of unknown electron densities with
$x_i = Ne_i$ equals the electron density in the voxel $i$, $h_{si}$ is the length of the ray path $s$ in the voxel $i$ and $r$ is the
vector of measurement errors assumed to be Gaussian distributed $r \sim N(0, R)$ with expectation 0 and covariance
matrix $R$.

### 2.2    Background model

As regularisation of the inverse problem in Eq. (2), a background model often provides the initial guess of the
state vector $x$. In this study, we apply the NeQuick model version 2.0.2. The NeQuick model was developed at the
International Centre for Theoretical Physics (ICTP) in Trieste/Italy and at the University of Graz/Austria (cf.
Hochegger et al., 2000; Radicella and Leitinger, 2001; Nava et al., 2008). The daily solar flux index F10.7 is used
to drive the NeQuick model.

### 2.3    Analysis step of the EnKF

We apply EnKF to solve the inverse problem defined in Section 2.1. Evensen (1994) introduces the EnKF as an
alternative to the standard Kalman Filter (KF) in order to cope with the non-linear propagation dynamics and the
large dimension of the state vector and its covariance matrix. In an EnKF, a collection of realisations, called
ensembles, represent the state vector $x$ and its distribution.
Let $X^f = [x_1^f, \dots, x_N^f]$ be a $(K \times N)$ matrix whose columns are the ensemble members, ideally following the a
priori distribution of the state vector $x$. Further, the observations collected in $y$ are treated as random
variables. Therefore, we define an $(m \times N)$ ensemble of observations $Y = [y_1, y_2, \dots, y_N]$ with $y_i = y + \epsilon_i$ and a
random vector $\epsilon_i$ from the normal distribution $N(0, R)$.
We define the ensemble covariance matrix around the ensemble mean $E(X^f) = \frac{1}{N}\sum_{j=1}^{N} x_j^f$ as follows:

$$P^f = \frac{1}{N-1}\sum_{j=1}^{N}\left\{\left(x_j^f - E(X^f)\right) \cdot \left(x_j^f - E(X^f)\right)^T\right\}. \tag{3}$$

In the analysis step of the EnKF, the a priori knowledge on the state vector $x$ and its covariance matrix is updated
by

$$X^a = X^f + P^f H^T (R + HP^f H^T)^{-1} \cdot (Y - HX^f), \tag{4}$$

where the matrix $X^a$ represents the a posteriori ensembles and hence the a posteriori state vector.
For the propagation of the analysed solution to the next time step, we test different propagation models described
in Section 2.4. In order to generate the initial ensembles $X^f(t_0)$ we use the NeQuick model and describe the
procedure in Section 2.5. Keeping in mind that we have to deal with an extremely large state vector (details are
presented in Section 3.1), the important advantage of the EnKF, for the present study, is that there is no need for
explicitly calculation of the ensemble covariance matrix (cf. Eq. (3)). Instead, to perform the analysis step in Eq.
(4) we follow the implementation suggested by Evensen (2003).

## 2.4 Considered models for the propagation step of the EnKF

In this section, we introduce different models to propagate the analysed solution to the next time step. With all of
them, we propagate the ensembles 20 minutes in time. Generally, these propagation models can be described as
$X^f(t_{n+1}) = F(X^a(t_n)) + W_F(t_{n+1}) + \Omega_F(t_{n+1})$. In the following subsections, we outline possible choices of the
model $F$, the systematic error $W_F$ and the process noise $\Omega_F$.
Note: Beyond the presented methods, in addition we had tested a propagation model based on "persistence", i.e.
$X^f(t_{n+1}) = X^a(t_n) + W_{persis}(t_{n+1}) + \Omega_{persis}(t_{n+1})$. Already after a time period of about 24 hours, this method
had shown unreasonable effects in the reconstructions, like a completely misplaced equatorial crest region.

### 2.4.1 Method 1: Rotation

The method Rotation assumes that in geomagnetic coordinates, the ionosphere remains invariant in space while
Earth rotates below it (cf. Angling and Cannon, 2004). Thus, we propagate the analysed ensemble $X^a(t_n)$ from
time $t_n$ to the next time step $t_{n+1}$ by:

$$X^f(t_{n+1}) = Rot(X^a(t_n)) + W_{rot}(t_{n+1}). \tag{5}$$

To calculate $Rot(X^a(t_n))$ the geomagnetic longitude is changed corresponding to the evolution time $\Delta t = t_{n+1} -$
$t_n$, i.e. 5 degree of longitude per 20 minutes. $W_{rot}$ denotes the systematic error introduced by approximation of
the true propagation of $X^f$ by a simple rotation. We tested here the following estimation of $W_{rot}$:

$$W_{rot}(t_{n+1}) = ratio_{rot}(t_{n+1}) \cdot E\left(Rot(X^a(t_n))\right) \cdot \epsilon_{1\times N} \text{ with} \tag{6}$$

$$ratio_{rot}(t_{n+1}) = \frac{\left(x^b(t_{n+1}) - Rot(x^b(t_n))\right)}{3 \cdot Rot(x^b(t_n))}, \tag{7}$$

where $x^b$ is the electron density vector calculated by the NeQuick model and $\epsilon_{1\times N}$ is an $(1 \times N)$ matrix of ones.
The division in the second equation is element-wise. The ratio $ratio_{rot}(t_{n+1})$ in Eq. (7) represents the relative
error introduced by the application of $Rot\left(x^b(t_n)\right)$ instead of $x^b(t_{n+1})$. In this way, we obtain in Eq. (6) an
approximation of the mean error introduced by approximation of the true state at time $t_{n+1}$ by the rotation of the
true state at time $t_n$. The factor $\frac{1}{3}$ has been chosen empirically as the result of an internal validation not presented
within this paper.

### 2.4.2 Method 2: Exponential decay


Here we assume the electron density differences between the voxels of the analysis and the background model to
be a first order Gauss-Markov sequence. These differences are propagated in time by an exponential decay function
(cf. Nikoukar et al. 2015, Bust and Mitchell, 2008; Gerzen et al., 2015)

$$X^f(t_{n+1}) = X^b(t_{n+1}) \cdot \epsilon_{1 \times N} + f(t_{n+1}) \cdot [X^a(t_n) - X^b(t_n)], \tag{8}$$

where $X^b(t)$ is the ensemble of electron density vectors calculated by the NeQuick model for the time $t$ as
described in Section 2.5; $f(t_{n+1}) = \exp\left(-\frac{\Delta t}{\tau}\right)$; $\Delta t = t_{n+1} - t_n$; $\tau$ denotes the temporal correlation parameter
chosen here as 3 hours.
Note: Similar to the method described here, we tested also the application of $Rot([X^a(t_n) - X^b(t_n)])$ instead of
$[X^a(t_n) - X^b(t_n)]$ in Eq. (8). The results were similar and are therefore not presented here.

### 2.4.3 Method 3: Rotation with exponential decay


For the third method, we define the propagation model as a combination of the propagation models described in
the previous subsections, in particular

$$X^f(t_{n+1}) = x^b(t_{n+1}) \cdot \epsilon_{1 \times N} + f(t_{n+1}) \cdot Rot([X^a(t_n) - x^b(t_n) \cdot \epsilon_{1 \times N}]) + W(t_{n+1}) + \sqrt{\frac{\Delta t}{20}} \cdot \Omega_{exp}(t_{n+1}). \tag{9}$$

The systematic error $W$ is estimated as

$$W(t_{n+1}) = f(t_{n+1}) \cdot \frac{8}{10} \cdot W_{rot}(t_{n+1}). \tag{10}$$

Thereby $f$ and $W_{rot}$ are defined as in the two previous sections. The factor $\frac{8}{10}$ thereby is again chosen empirically.
The process noise $\Omega_{exp}$ is assumed to be white with $\Omega_{exp}(t_{n+1}) = f(t_{n+1}) \cdot \Omega_{rot}(t_{n+1}) + \left(1 - f(t_{n+1})\right) \cdot$
$Q_{exp}(t_{n+1})$. Here the matrix $\Omega_{rot}$ consists of random realizations of the distribution $N(0, \Sigma^{rot})$ with

$$\Sigma_{ii}^{rot}(t_{n+1}) = \left(ratio_i \cdot \left\{E\left(Rot(X^a(t_n))\right)\right\}_i\right)^2, \tag{11}$$

where $ratio_i$ increases continuously depending on the altitude of the voxel $i$ from $\frac{0.5}{100}$ for lower altitudes to $\frac{1}{100}$ for
the higher altitudes (chosen empirically); $E\left(Rot(X^a(t_n))\right)$ denotes the ensemble mean vector. The equations (9)
and (11) can be interpreted as follows: For the chosen time step of 20 minutes, the standard deviation of the time
model error regarding the voxel $i$ is equal to $\sqrt{\Sigma_{ii}^{rot}(t_{n+1})} = ratio_i \cdot \left\{E\left(Rot(X^a(t_n))\right)\right\}_i$, varying between 0.5%
and 1% of the corresponding analysed electron density in the voxel $i$. In detail, we generate at each time step a
new vector $\rho_i \sim N(0,1)$ with $\dim(\rho_i) = 100 \times 1$ and calculate the $i$-th row $\omega_i^{rot}$ of $\Omega_{rot}$ by Eq. **Fehler!**
**Verweisquelle konnte nicht gefunden werden.**.

$$\omega_i^{rot}(t_{n+1}) = \sqrt{\Sigma_{ii}(\Omega_{rot}(t_{n+1}))} \cdot \rho_i(t_{n+1})^T. \tag{12}$$

The matrix $Q_{exp}(t_{n+1})$ consists of random realizations (different for each time step) consistent with the a priori
covariance matrix $L$ of the errors of the background $x^b(t_{n+1})$ (cf. Howe and Runciman, 1998). In detail: The a
priori covariance is assumed to be diagonal and $L_{ii}$ equals the square of 1% of the corresponding background
model value. Then the $i$-th row of $Q_{exp}$ is calculated by Eq. (13):

$$q_i(t_{n+1}) = \sqrt{L_{ii}(t_{n+1})} \cdot \rho_i(t_{n+1})^T. \tag{13}$$

## 2.5  Generation of the ensembles
In order to generate the ensembles we vary the F10.7 input parameter of the NeQuick model (cf. Section 2.2).
First, we analysed the sensitivity of the NeQuick model on F10.7. Based on the results, we calculate a vector
$\boldsymbol{F10.7}(t)$ of the solar radio flux index with $\dim(\boldsymbol{F10.7}(t)) = 100 \times 1$ and $\boldsymbol{F10.7}(t) \sim N\left(\text{F10.7}(t), \frac{3}{100} \cdot\right.$
$\left.\text{F10.7}(t)\right)$ at time $t$. The vector $\boldsymbol{F10.7}$ serves as input for the NeQuick model to calculate the 100 ensembles of
$X^b$ during the considered period and the initial guess of the electron densities $X^f(t_0)$.
An example on the variation of the generated ensembles is provided by **Figure 1**. Particularly, we show in this
figure the distribution of the differences between the ensemble of electron densities $X^b(t)$ and the NeQuick model
values for DOYs 041 and 076. The residuals are depicted for a selected altitude and chosen UT times, presented
through different colors (cf. subfigure history). In addition, the mean, the standard deviation (STD) and the root
mean square (RMS) of the residuals are presented in the subplots.
## 2.6  Provision of a benchmark by SMART+
In order to provide a benchmark for the described methods, we apply SMART+ as an additional reconstruction
technique. SMART+ is a combination of an iterative simultaneous multiplicative column normalized method
SMART (cf. Gerzen and Minkwitz, 2016) and a 3D successive correction method (3D SCM) (cf. e.g. Kalnay,
2011; Gerzen and Minkwitz, 2016). As first step, SMART distributes the STEC measurements among the electron
densities in the ray-path intersected voxels. For a voxel $i$, the multiplicative innovation is calculated as a weighted
mean of the ratios between the actual measurements and the currently expected measurements. The weights are
given by the length of the ray path corresponding to the measurement in the voxel $i$ divided by the sum of lengths
of all rays crossing the voxel $i$. Consequently, only voxels intersected by at least one measurement are innovated
during the SMART procedure. Thereafter, assuming non-zero correlations between the ray path intersected voxels
and those not intersected by any STEC, an extrapolation is done from intersected to not intersected voxels. For
this purpose, one iteration of the 3D SCM is applied. For more details we refer to Gerzen and Minkwitz (2016)
and Gerzen et al. (2017).
For SMART+ the number of iterations at each time step is set to 25 and the correlation coefficients are chosen as
described in Gerzen and Minkwitz (2016). For each time step, SMART+ reconstructs the electron densities based
on the background model (here NeQuick) and the currently available measurements. In other words, there is no
propagation of the estimated electron densities from a time step $t_n$ to the time step $t_{n+1}$.

## 3    Validation scenario

Within this study, the EnKF with the different propagation methods is applied and validated for the tomography of the topside ionosphere and plasmasphere. Two periods with quiet (DOY 041-059, 2015) and perturbed (DOY 074-079, 2015) ionospheric conditions are analysed. In this scope, we investigate the ability to reproduce assimilated STEC as well as to estimate independent STEC measurements and in-situ electron density measurements of the Swarm Langmuir Probes (LP). In addition, we apply the tomography approach SMART+ (cf. Section 2.6) to provide a benchmark.

### 3.1    Reconstruction area

We estimate the electron density over the entire globe with a spatial resolution of 2.5 degrees in geodetic latitude and longitude. Altitudes between 430 km and 20 200 km are reconstructed where the resolution equals 30 km for altitudes from 430 km to 1000 km and decreases exponentially with increasing altitude for altitudes above 1000 km, i.e. in total 42 altitudes. Consequently, the number of unknowns is $K = 217728$. The temporal resolution $\Delta t$ is set to 20 minutes.

### 3.2    Ionospheric conditions in the considered periods

We use the solar radio flux F10.7, the global planetary 3h index Kp and the geomagnetic disturbance storm time (DST) index to characterize the ionospheric conditions during the periods of DOY 041-059 and DOY 074-079 2015. In the February period (DOY 041-059, 2015) the ionosphere is evaluated as quiet with F10.7 between 108 and 137 sfu, a Kp index below 6 (on two days between 4 and 6, during the rest of the period below 4) and DST values between 20 and -60 nT. The 17-th of March (DOY 076) 2015 is known as the St. Patrick's Day storm. The F10.7 value equals ~116 sfu on DOY 075 and ~113 sfu on DOY076, the Kp index is below 5 on DOY 075 and increases to 8 on DOY 076; DST drops down to -200nT on DOY 076.

### 3.3    Data

#### 3.3.1    STEC measurements

As input for the tomography approaches and for the validation, we use space-based calibrated STEC measurements of the following LEO satellite missions: COSMIC, Swarm, TerraSAR-X, MetOpA and MetOpB, and GRACE. Please note that in 2015, the orbit height of the COSMIC and MetOp satellites is ~800 km, the orbit height of the Swarm B and TerraSAR-X satellites is ~500 km and the one of the Swarm C satellite ~460 km. The STEC measurements of Swarm A and GRACE are used for the validation only. The Swarm A satellite flew side by side with the Swarm C satellite at around 460 km height. The height of the GRACE orbit was around 430 km. All satellites flew at almost polar orbits. More information about the LEO satellites may be found on the following webpages:

COSMIC: https://www.nasa.gov/directorates/heo/scan/services/missions/earth/COSMIC.html),

Swarm: (https://www.esa.int/Applications/Observing_the_Earth/Swarm),

TerraSAR-X: (https://earth.esa.int/web/eoportal/satellite-missions/t/terrasar-x),

MetOpA and MetOpB: (https://directory.eoportal.org/web/eoportal/satellite-missions/m/metop),

GRACE: (https://www.nasa.gov/mission_pages/Grace/index.html).

The STEC measurements of the Swarm satellites are acquired from https://swarm-diss.eo.esa.int/ and the STEC
measurements of the other satellite missions are downloaded from http://cdaac-
www.cosmic.ucar.edu/cdaac/tar/rest.html. Both data providers supply also information on the accuracy of the
STEC data. We utilize this information to fill the covariance matrix $R$ of the measurement errors. The collected
STEC data is checked for plausibility before the assimilation.

### 3.3.2 In-situ electron density measurements from the Swarm Langmuir Probes

The LPs on board the Swarm satellites provide in-situ electron density measurements with a time resolution of 2
Hz. For the present study, the LP in-situ data are acquired from https://swarm-diss.eo.esa.int/. In addition, further
information on the pre-processing of the LP data is made available on this website.
Lomidze et. al (2018) assess the accuracy and reliability of the LP data (December 2013 to June 2016) by nearly
coincident measurements from low- and middle-latitude incoherent scatter radars, low-latitude ionosondes, and
COSMIC satellites, which cover all latitudes. The comparison results for each Swarm satellite are consistent across
these different measurement techniques. The results show that the Swarm LPs underestimate the electron density
systematically by about 10%.

## 4 Results

In this section, the different methods are presented with the following color code: blue for the method Rotation,
green for the method Exponential decay, light blue for the method Rotation with exponential decay, magenta for
NeQuick and red for SMART+. The legends in the figures are the following: "Rot" for the method Rotation, "Exp"
for the method Exponential decay, "Rot and Exp" for the method Rotation with exponential decay.

### 4.1 Reconstructed electron densities

At the end of each EnKF analysis step, we have, for each of the considered methods, 100 ensembles representing
the electron density values within the voxels. The EnKF reconstructed electron densities are then calculated as the
ensemble mean. The top subplots of **Figure 2** present the electron densities reconstructed by the method Rotation
with exponential decay, i.e. $E\left(X_{Rot\ and\ Exp}^a(t_n)\right)$, for $t_n$ corresponding to DOY 076, at 19:00 UT. The upper left
corner subplot shows horizontal layers of the topside ionosphere at different heights between 490 and 827 km. The
subplot in the upper right corner illustrates the plasmasphere for altitudes between 827 and 2400 km at selected
longitudes. The bottom line subplots show VTEC maps deduced from the 3D electron density in the considered
altitude range between 430 and 20200 km, where the left hand side subplot represents the reconstructed values and
the right hand side VTEC calculated from the NeQuick model. It is observed that the reconstructed VTEC values
are slightly higher than the ones of the NeQuick model.
**Figure 3** displays the electron density layers reconstructed by the method Rotation, i.e. $E\left(X_{Rot}^a(t_n)\right)$, for $t_n$
corresponding to DOY 076, at 19:00 UT. Again, reconstructed electron densities at heights between 490 and 827
km (left) and the corresponding VTEC map deduced from the reconstructed 3D electron density (right) are
depicted. All reconstructed values seem to be plausible, showing as expected the crest region, low electron
densities in the Polar regions, etc. The method Rotation delivers much higher values than the NeQuick model, cf.
**Figure 2**. In **Figure 4**, we take a closer look at the differences between the modelled and reconstructed electron
densities.

In the following, we discuss **Figure 4** - **Figure 7**, in order to understand the deviations between the reconstructions produced by the different methods. On **Figure 4** the differences between the reconstructed and the modelled electron densities, i.e. $E(X^a(t_n)) - x^b(t_n)$, are shown for all methods: Rotation with exponential decay, Rotation, Exponential decay and SMART+ (from top left subfigure to bottom right subfigure) on DOY 076 at 19:00 UT. In addition, **Figure 5** expresses these differences in percent. Please note the different ranges of the colorbars for the subfigures. **Figure 6** illustrates the orbits of the LEO satellites for the STEC measurements used for the reconstructions on DOY 076, at 19:00 UT (left) and the corresponding ground-track (right). The highest differences are observed for the methods Rotation and Exponential decay, whereas the method Rotation with exponential decay yields the smallest differences. Furthermore, as expected, the EnKF approaches provide smooth and coherent patterns of differences in the ionization. Contrary, the complementary approach of SMART+ has rather small patterns in areas where measurements are available and falls back to the background model in areas without measurements in the surrounding. In this context, the correlation lengths between the electron densities are of importance. These correlation lengths are set empirically in SMART+, whereas EnKF establishes them automatically, i.e. without setting or estimating them explicitly as for instance in SMART+ or Kriging approaches. For a comprehensive evaluation of the quality of the different reconstructions in the context of the used correlation lengths, future analyses with further validation data and in dependence on the coincidences between the measurement geometry and the geometry of the validation data set are necessary.

Taking into account the differences in **Figure 5**, for instance around 120°E, and the measurement geometry in **Figure 6**, it is evident that the estimates of the EnKF are not only based on the current measurements but also on a priori information obtained from assimilations before DOY 076, 2015, 19:00 UT. This is of course not the case for SMART+.

In order to supplement the understanding on the differences between the propagation methods, **Figure 7** presents the differences $E\left(X^f_{method}(t_{n+1})\right) - E\left(X^a_{method}(t_n)\right)$ on the left column subfigures; and the percentage differences $100 \cdot \left[E\left(X^f_{method}(t_{n+1})\right) - E\left(X^a_{method}(t_n)\right)\right] \Big/ \frac{1}{2} \cdot \left[E\left(X^f_{method}(t_{n+1})\right) + E\left(X^a_{method}(t_n)\right)\right]$ in the right column for $t_n$ corresponding to DOY 076, at 19:00 UT. Particularly, the methods (from top to bottom): Rotation with exponential decay, Rotation and Exponential decay are presented. The differences for the methods Rotation and Rotation with exponential decay clearly indicate the rotation of the crest region (cf. also **Figure 3**). The method Rotation with exponential decay works less rigorously in the rotation than the method Rotation since it is anchored by the background model and the rotation of the differences $X^a(t_n) - x^b(t_n)$ is damped by the exponential decay function, see Eq. (9). Contrary to these two methods, the method Exponential decay tries to propagate the difference $X^a(t_n) - X^b(t_n)$ to the next time step and adds them to the background $X^b(t_{n+1})$. Hence, we observe in the lower left corner subplot of **Figure 7** a similar pattern as in the corresponding lower left corner subplot of **Figure 4**.

Concluding, the different behaviour of the propagation methods in combination with the sparse measurement geometry might serve as an explanation for the substantial differences observed in the VTEC maps shown in **Figure 2** and **Figure 3**.

## 4.2    Plausibility check by comparison with assimilated STEC

In this Section, we check the ability of the methods to reproduce the assimilated STEC measurements. For that purpose, we calculate STEC along a ray path $j$, for all ray path geometries, using the estimated 3D electron densities, denoted as $STEC_j^{est}$, and compare them with the measured STEC, $STEC_j^{meas}$, used for the reconstruction. Then the mean deviation $\Delta \boldsymbol{STEC}$ between the measurements $STEC_j^{meas}$ and the estimate $STEC_j^{est}$ is calculated for each of the considered methods according to

$$\Delta \boldsymbol{STEC}(t_n) = \frac{1}{m} \sum_{j=1}^{m} \left( |STEC_j^{meas}(t_n) - STEC_j^{est}(t_n)| \right), \tag{14}$$

where $m$ = number  of assimilated measurements. $\Delta \boldsymbol{STEC}$ is calculated at each epoch $t_n$. In terms of the notation used for the Eqs. (1) - (4), we can reformulate the above formula for the mean deviation as

$$\Delta \boldsymbol{STEC}(t_n) = \frac{1}{m} \sum_{j=1}^{m} \left( |y_j(t_n) - E(X_a(t_n))^T \cdot H_j| \right), \text{ with } H_j = j\text{-th row of } H. \tag{15}$$

Further, we consider the **RMS** of the deviations, in detail

$$\boldsymbol{RMS}(t_n) = \sqrt{\frac{1}{m} \sum_{j=1}^{m} \left( |STEC_j^{meas}(t_n) - STEC_j^{est}(t_n)| \right)^2}. \tag{16}$$

To calculate $\Delta \boldsymbol{STEC}$ and $\boldsymbol{RMS}$, the same measurements are used as for the reconstruction. In this sense, the results presented in **Figure 8** - **Figure 12**  serves as a plausibility check, testing the ability of the methods to reproduce the assimilated TEC.

**Figure 8** depicts the distribution of the residuals, left subfigure for the quiet period, right subfigure for the perturbed period. The corresponding residual median, standard deviation (STD) and root mean square (RMS) values are also presented in the figure. It is worth to mention here that during the quiet period, the measured STEC is below 150 TECU. For all DOYs of the perturbed period, except DOY 076, the measured STEC is below ~130 TECU. On DOY 076, the STEC values rise up to 370 TECU.

The NeQuick model seems to underestimate the measured topside ionosphere and plasmasphere STEC during both periods. During both periods, SMART+ seems to perform best, followed by the method Rotation. However, Rotation produces higher STD and RMS values. Compared to the NeQuick residuals, SMART+ is able to reduce the median of the residuals by up to 86% during the perturbed and up to 79% during the quiet period. The RMS is reduced by up to 48% and the STD by up to 41%. Rotation reduces the NeQuick median by up to 83%, the RMS by up to 27%, the STD value is almost on the same level as for NeQuick. The method Exponential decay is able to decrease the median of the NeQuick residuals by up to 54%, the RMS by up to 25%, and the STD values by up to 13%. The method Rotation with exponential decay performs similar to the NeQuick model.  The latter could indicate that the parameters chosen for the error terms and weighting in Eq. (9) could still be improved, although an extensive validation of these parameters was performed prior to the analyses presented in this paper and the best configuration was selected.

Interestingly, the median values are higher during the quiet period, while the STD values are on the same level compared between perturbed and quiet periods. The reason therefore is probably that the assimilated STEC values have in average lower magnitude during the days in the perturbed period, compared to those during the quiet period

(which explains the lower median), except the storm DOY 076, while on DOY 076 they are significantly higher
(which explains the comparable STD ).
**Figure 9** and **Figure 10** plot $\Delta \boldsymbol{STEC}$ values versus time for the selected periods. Noticeable is the increase of
$\Delta \boldsymbol{STEC}$ during the storm on DOY 76. On the rest of the period, $\Delta \boldsymbol{STEC}$ is below eight TECU. During both periods,
SMART+ generates the lowest $\Delta \boldsymbol{STEC}$ values. $\Delta \boldsymbol{STEC}$ of the methods Rotation and Exponential decay are in most
of the cases higher than SMART+ delta STEC values and lower than the NeQuick model. $\Delta \boldsymbol{STEC}$ of the method
Rotation with exponential decay is similar to the NeQuick model.
**Figure 11** and **Figure 12** present the distribution of $\Delta \boldsymbol{STEC}$ and the $\boldsymbol{RMS}$ error (cf. Eq. (15)) for the quiet and
perturbed periods respectively. **Figure 11** confirms the conclusions we draw so far from **Figure 8** and **Figure 9** .
SMART+ delivers the lowest $\Delta \boldsymbol{STEC}$ and $\boldsymbol{RMS}$ values, followed by the method Rotation and then by the method
Exponential decay. Rotation with exponential decay performs similar to the NeQuick model. For the perturbed
period, again SMART+ delivers the lowest $\Delta \boldsymbol{STEC}$ and $\boldsymbol{RMS}$ statistics, followed by the Exponential decay and
the Rotation with similar results.
### 4.3 Validation with independent space-based sTEC data
In order to validate the methods with respect to their capability to estimate independent STEC, the LEO satellites
Swarm A and GRACE have been used. The STEC measurements of these satellites are not assimilated by the
tested methods.
For each of the three LEOs, the residuals between $STEC^{meas}$ and $STEC^{est}$ are calculated and denoted as $dTEC =$
$STEC^{meas} - STEC^{est}$. Further, the absolute values of the residuals $|dTEC|$ are considered.
In general, for the quiet period, the STEC measurements of Swarm A vary below 105 TECU and for the second
period below 170 TECU. For the GRACE satellite, the STEC measurements are below 282 TECU for the quiet
period and below 264 TECU for the second period.
**Figure 13** and **Figure 14** display the histograms of the STEC residuals during the quiet period for Swarm A and
GRACE respectively. Presented are the distributions of the residuals $dTEC$ and the absolute residuals $|dTEC|$.
Also plotted are the median, STD and RMS of the corresponding residuals. **Figure 15** and **Figure 16** depict the
histograms of the STEC residuals during the perturbed period.
Again, the NeQuick model seems to underestimate the measured STEC during both periods for GRACE and
Swarm A satellites. Compared to the NeQuick model, during both periods, the methods SMART+ and Exponential
decay decrease the residuals and the absolute residuals between measured and estimated STEC for both GRACE
and Swarm A satellites. The method Rotation with exponential decay performs for both periods very similar to the
NeQuick model. The performance of the method Rotation is partly even worse than the one of the background
model. Our impression is that the number and the distribution of the assimilated measurements is too small and
angle limited to be sufficient to dispense with a background model, as is the case with the Rotation method, which
uses the model only for the estimation of the systematic error.
Regarding the STEC of Swarm A, the lowest residuals and the most reduction in comparison to the NeQuick
model, are achieved by SMART+. The median and the STD of the SMART+ residuals are ~0.3 TECU and ~3.4
TECU respectively for quiet and ~ 0.7 TECU and ~7 TECU for the perturbed period. Compared to the NeQuick
model, the absolute median value is reduced up to 64% by SMART+ during the quiet and by up to 61% during the
perturbed period. The STD value is decreased by up to 47% during the quiet and up to 29% during the storm
period. The second lowest residuals are achieved by the Exponential decay - here the median of the residuals is
around 0.2 TECU for quiet and around 0.8 TECU for the perturbed period.
Regarding the STEC of GRACE during the quiet period, the lowest residuals and the most reduction in comparison
to background, are achieved by the Exponential decay, followed by SMART+. Exponential decay reduces the
background absolute median value by up to 26% and the STD value by up to 28%. The median of the residuals is
around 0.2 TECU. For SMART+, the median of the residuals is around 2.9 TECU. During the perturbed period,
SMART+ reduces the absolute median at most by 17% and the STD by 9%, the Exponential decay does not reduce
the absolute median, compared to NeQuick, but it reduces the absolute STD value by 23%. The median of the
residuals are around -0.5 TECU for Exponential decay and around 0.8 TECU for SMART+.
Comparing between quiet and storm conditions, in general an increase of RMS and STD of the SWARM A
residuals is observable for the NeQuick model and all tomography methods regarding both satellites. This is not
the case for the GRACE residuals.

### 4.4 Validation with independent LP in-situ electron densities

In this section, we further extend our analyses to the validation of the methods with independent LP in-situ electron
densities of the three Swarm satellites. According to the locations of the LP measurements, the estimated electron
density values are interpolated (by a 3D interpolation, using the MATLAB build-in function
scatteredInterpolan.m) from the 3D electron density reconstructions. For each satellite, the measured electron
density $Ne^{meas}$ is compared to the estimated one $Ne^{est}$. In particular we calculate the residuals $dNe = Ne^{meas} -$
$Ne^{est}$, the absolute residuals $|dNe|$, the relative residuals $dNe_{rel} = \frac{dNe}{Ne^{meas}} \cdot 100\%$ and the absolute relative
residuals $|dNe_{rel}|$.
**Figure 17** depicts the distribution of the residuals $dNe$ for the quiet period along with the median, STD and RMS
values. Each of the three subplots presents one of the Swarm satellites. In **Figure 18** the histograms of $|dNe|$ and
$|dNe_{rel}|$ are given for the same period. In **Figure 18** we do not separate the values for the different satellites,
because these are similar. **Figure 19** and **Figure 20** show the corresponding histograms for the perturbed period.
The electron densities of the NeQuick model are in median slightly higher than the LP in-situ measurements for
all three satellites during both periods. The median and STD values for the $|dNe_{rel}|$ residuals produced by
NeQuick are ~33% and ~38% resp. during the quiet period. For the perturbed period, we observe higher median
and STD values of ~45% and ~56%, resp. The increase of the RMS and STD values of the absolute residuals is
also visible for all the considered reconstruction methods.
The methods SMART+ and Rotation with exponential decay follow the trend of the model and show similar
distributions in **Figure 17** and **Figure 19**. Comparing these two methods with the NeQuick model, the performance
of SMART+ is slightly better reducing the median of the absolute and absolute relative residuals by up to 8%.
Further, during both periods, SMART+ reduces the STD values of the $|dNe|$ values by up to 23%. However, the
STD and RMS values of the $|dNe_{rel}|$ residuals for SMART+ during the quiet period are higher than those of the
NeQuick model. The median and STD values of the $|dNe_{rel}|$ residuals for SMART+ are ~30% and ~43% resp.
during quiet and higher during perturbed period, namely ~43% and ~53% resp. The statistics of the methods
Exponential decay and Rotation are worse than those of NeQuick.

## 5    Summary and conclusions

In this paper, we assess three different propagation methods for an Ensemble Kalman Filter approach in the case that a physical propagation model is not available or discarded due to computational burden. We validate these methods with independent STEC observations of the satellites GRACE and Swarm A and with independent Langmuir probes data of the three Swarm satellites. The methods are compared to the algebraic reconstruction method SMART+, serving as a benchmark and to the background model NeQuick for periods of the year 2015 covering quiet to perturbed ionospheric conditions.

Overlooking all the validation results, the methods SMART+ and Exponential decay reveal the best performance with the lowest residuals, whereas the method Rotation with exponential decay provides only a small improvement compared to the NeQuick model. While SMART+ modifies the electron densities of the background model around the measurement geometry and produces rather small patches, the EnKF produces larger and smoother patterns. As expected, the validations indicate that the electron density estimates of the EnKF are not only dependent on the current measurement geometry but also on prior assimilations.

The plausibility check in section 4.2 shows that all methods reduce successfully the STEC residuals and provide better results than the background model. SMART+ demonstrates the best performance and lowers the error statistics of the NeQuick model by up to 86%, followed by the method Rotation, decreasing the median of the residuals by up to 83%. The method Exponential decay reduces the median by up to 55%, but the STD values stay almost on the same level as for the NeQuick model.

Although the EnKF with the method Rotation reproduces the assimilated STEC data well, less accurate estimates are obtained in the validation with independent data. We assume this has two main reasons: First, as the only propagation method, Rotation is not anchored by the background model. Second, the number of the assimilated measurements is low compared to the number of unknowns and the available measurements are unevenly distributed and angle limited. Both together could lead to increased deviations of the estimates from the truth.

The methods SMART+ and the EnKF with Exponential decay provide the best estimates of the independent STEC and reduce the STEC residuals by up to 64% for Swarm A and 28% for GRACE, compared to the NeQuick model. SMART+ generates the smallest residuals for the STEC measurements of Swarm A and Exponential decay performs at best for STEC measurements of GRACE.

Concerning the estimation of independent electron densities of the Langmuir Probes, SMART+ shows the best results, reducing the absolute residuals by up to 23%. The median and STD values of the absolute residuals $|dNe_{rel}|$ for SMART+ are ~30% and ~43% respectively during quiet ionospheric conditions and ~43% and ~53% respectively during perturbed ionospheric conditions. The distributions of the residuals produced by Rotation with exponential decay are similar to the ones of the NeQuick model. In general, all the considered methods generate relatively high residuals. These observations could be explained by the fact that the independent electron density measurements are located at the lower edge of the reconstructed area and all the assimilated measurements are located above. Additionally, Swarm LPs was found to underestimate the true electron density systematically, cf. Section 3.3.2. In order to obtain better results for the lower altitudes, it might therefore be necessary to apply a kind of anchor point for the lower altitudes within the reconstruction procedure which could for instance be the Swarm LPs electron density measurements themselves.

Another approach to improve the reconstructions could be to precondition the background model, e.g. in terms of
F2 layer characteristics or the plasmapause location (cf. e.g. Bidaine and Warnant, 2010, Gerzen et. al., 2017).
To get a comprehensive final impression of the performance of the investigated methods and to gain insight into
the ability of the methods to correctly characterize the shapes of the electron density profiles, we intend to continue
the validation of the methods with additional independent measurements of the plasmasphere and topside
ionosphere, e.g. coherent scatter radar data.

**Acknowledgements**

We thank the NOAA (ftp://ftp.ngdc.noaa.gov/STP/GEOMAGNETIC_DATA/INDICES/) and WDC Kyoto
(http://wdc.kugi.kyoto-u.ac.jp/dstdir/index.html) for making available the geo-related data, F10.7, Kp and DST
indices. We are grateful to the European Space Agency for providing the Swarm data (https://swarm-
diss.eo.esa.int/) and to the CDAAC: COSMIC Data Analysis and Archive Center for providing the STEC data of
several LEO satellites (http://cdaac-www.cosmic.ucar.edu/cdaac/tar/rest.html). Additionally, we express our
gratitude to the Aeronomy and Radiopropagation Laboratory of the Abdus Salam International Centre for
Theoretical Physics Trieste/Italy for providing the NeQuick version 2.0.2 for scientific purposes. This study was
performed as part of the MuSE project (https://gepris.dfg.de/gepris/projekt/273481272?language=en), funded by
the DFG as a part of the Priority Programme "DynamicEarth", SPP-1788.

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

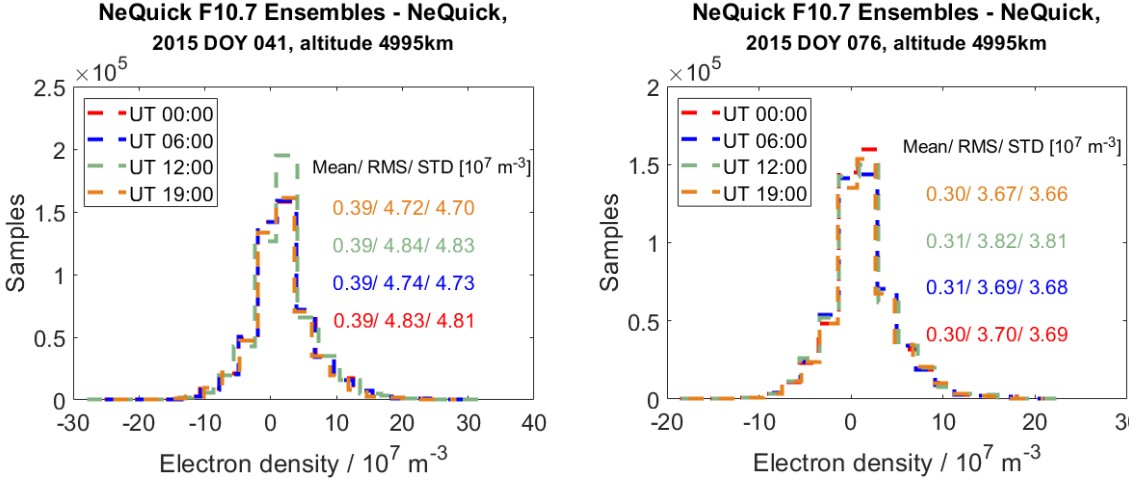

**Figure 1: The distribution of the ensemble residuals for a chosen altitude and selected UT times, for all latitudes, longitudes. Left: for DOY 041, right: for DOY 076.**

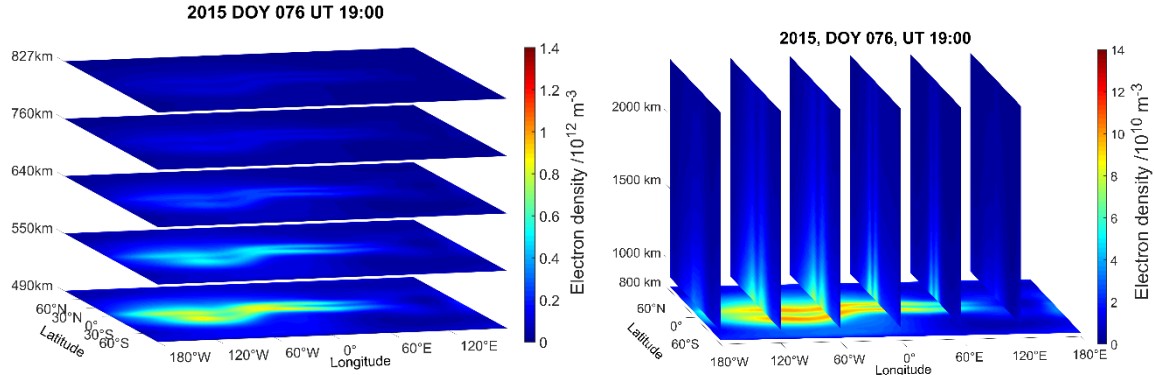

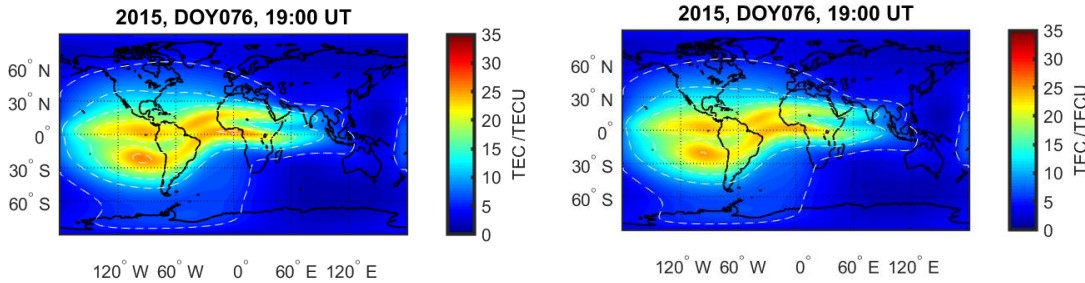

579

**Figure 2: Subfigures top: Rotation with exponential decay reconstructed electron density represented by layers at different heights between 490 and 827 km (left) and at chosen longitudes for altitudes between 827 and 2400 km (right). Subfigures bottom: The vertical TEC map deduced from the reconstructed (left) and NeQuick-modeled (right) 3D electron density in the altitude range between 450 and 20200 km.**

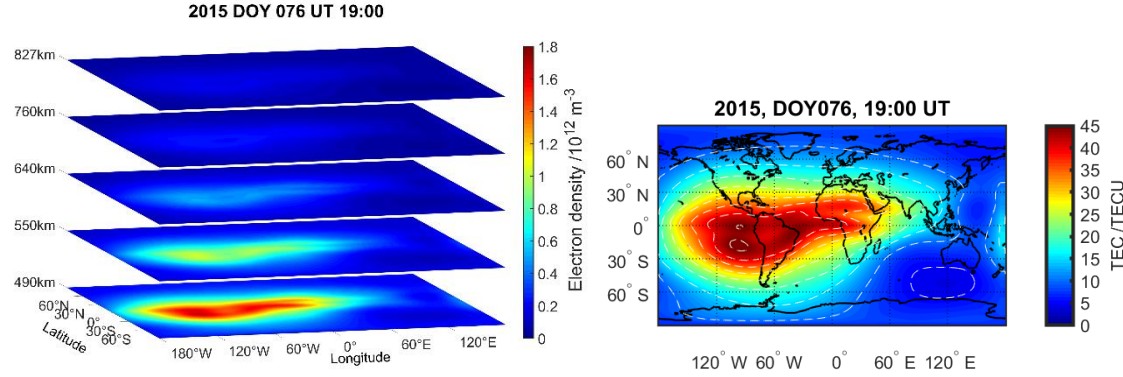

584

**Figure 3: Method Rotation reconstructed electron density represented by layers at different heights between 490 and 827 km (left) and vertical TEC map deduced from the reconstructed 3D electron density in the altitude range between 450 and 20200 km (right).**

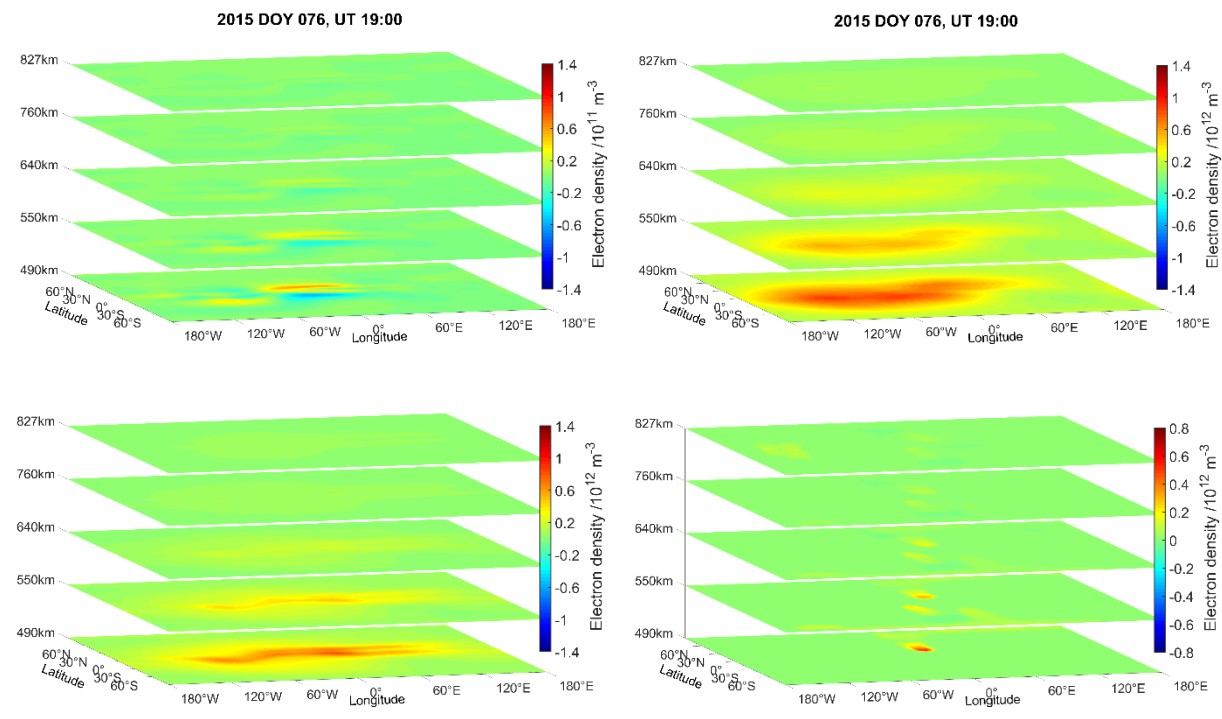

589

**Figure 4: Reconstructed minus NeQuick modeled electron density represented by layers at different heights between 490 and 827 km. Left top: For Rotation with exponential decay. Right top: Rotation. Left bottom: Exponential decay. Right bottom: SMART+.**

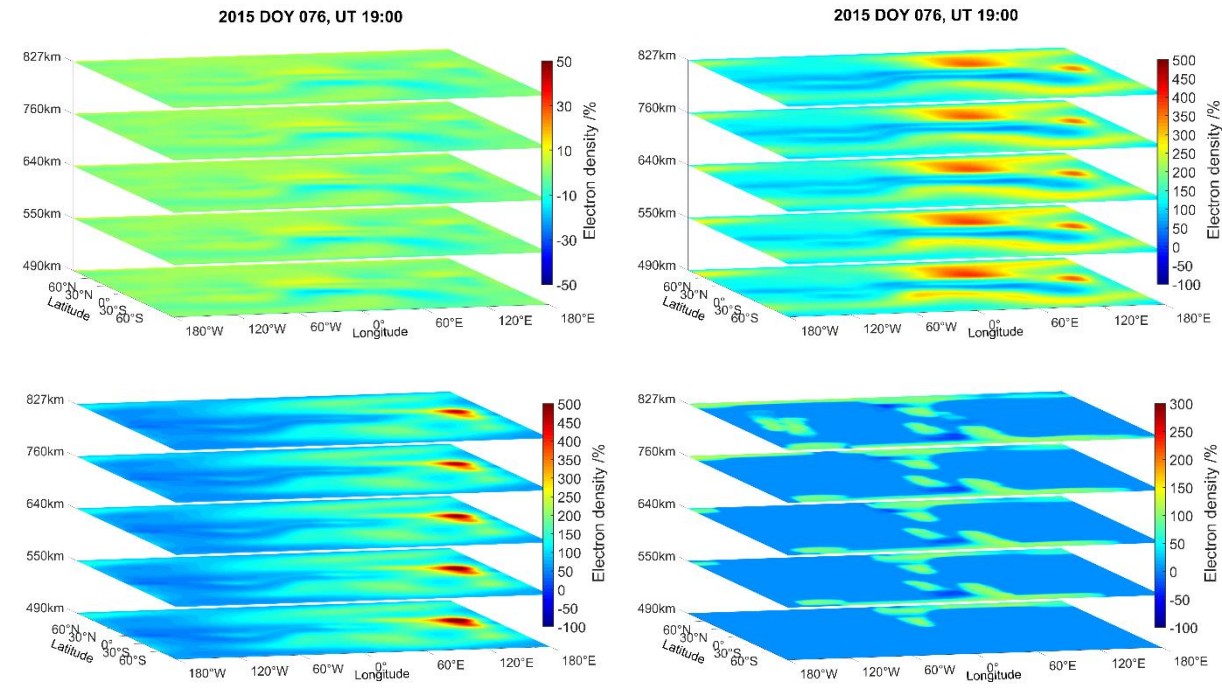

**Figure 5: Differences between reconstructed and NeQuick modeled electron density in percent, represented by layers at different heights between 490 and 827 km. Left top: For Rotation with exponential decay. Right top: Rotation. Left bottom: Exponential decay. Right bottom: SMART+.**

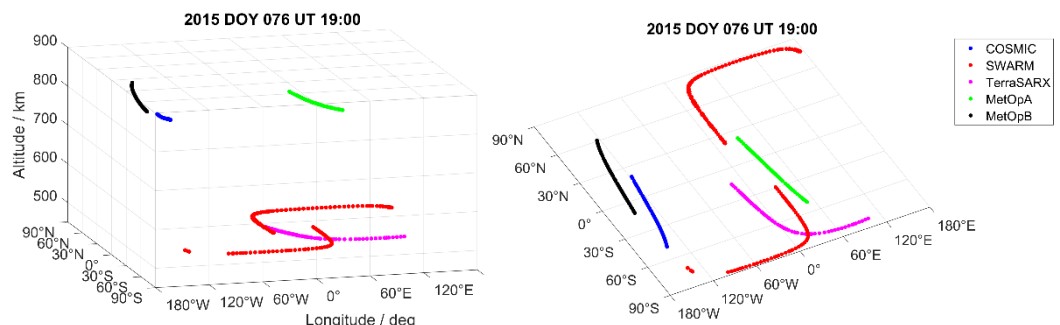

**Figure 6: The locations of the LEO satellites of the STEC measurements used for the reconstruction.**

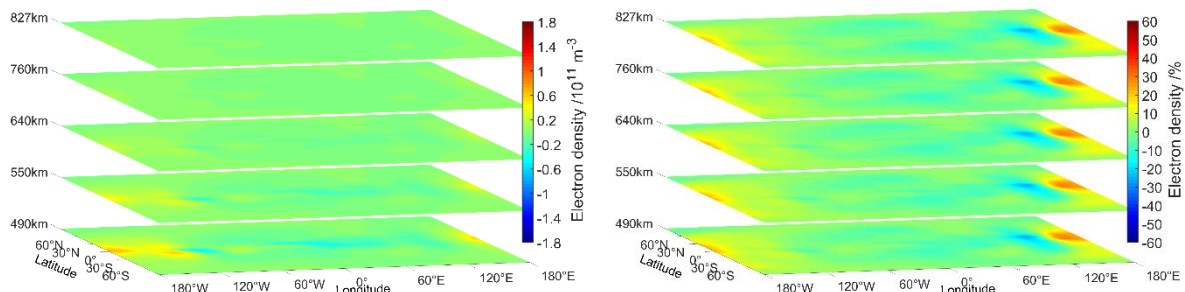

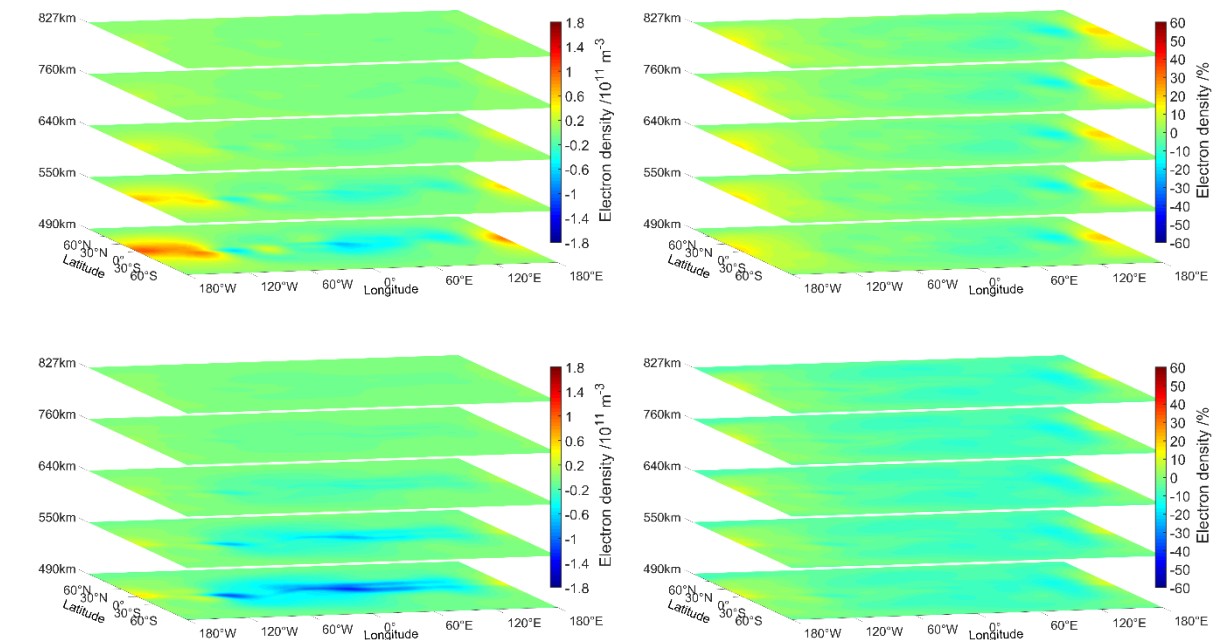



**Figure 7: Left: Differences between the forecasted and analysed electron densities, represented by layers at different heights between 490 and 827 km. Right: Differences in percent. Top: Method Rotation with exponential decay. Middle: Rotation. Bottom: Exponential decay.**

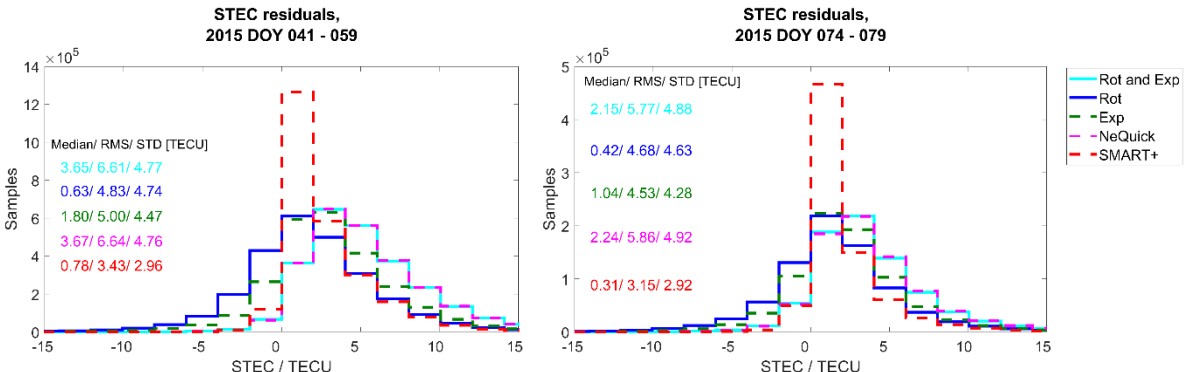


**Figure 8: Plausibility check – distributions of the STEC measured minus STEC estimated residuals. Left subfigure depicts residuals of the quiet period, right subfigure for the perturbed period.**

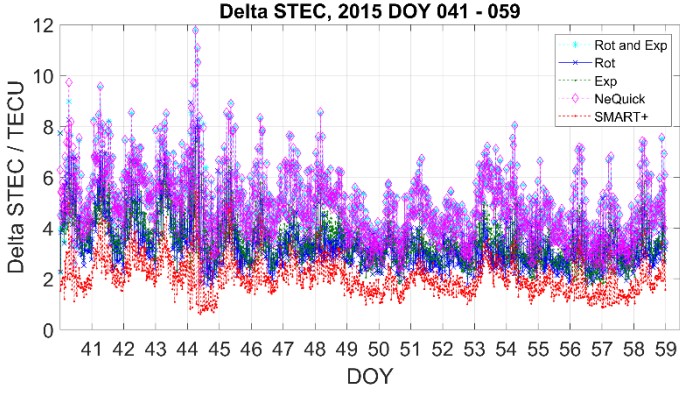


**Figure 9: Plausibility check for the quiet period – $\Delta STEC$ values versus time.**

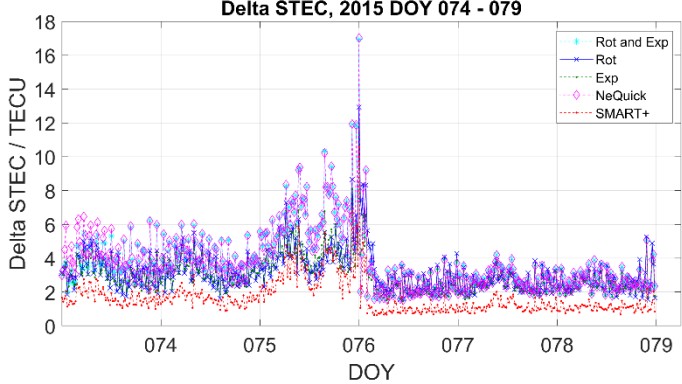


**Figure 10: Plausibility check for the perturbed period – $\Delta STEC$ values versus time.**

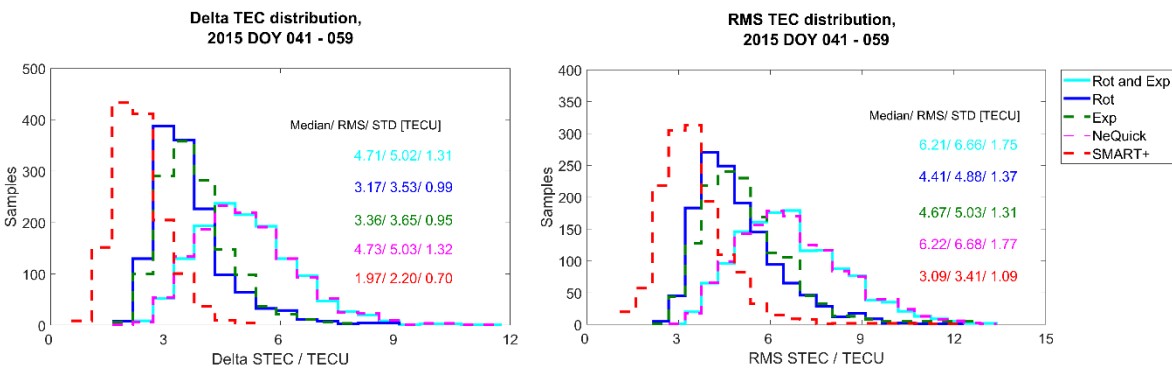


**Figure 11: Plausibility check for the quiet period – distributions of the delta TEC (left) and RMS (right) values.**

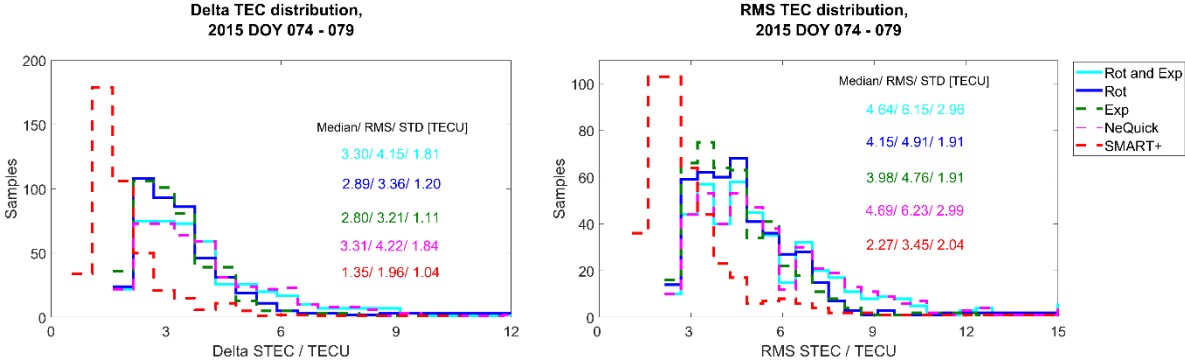


**Figure 12: Plausibility check for the perturbed period – distributions of the delta TEC (left) and RMS (right) values.**

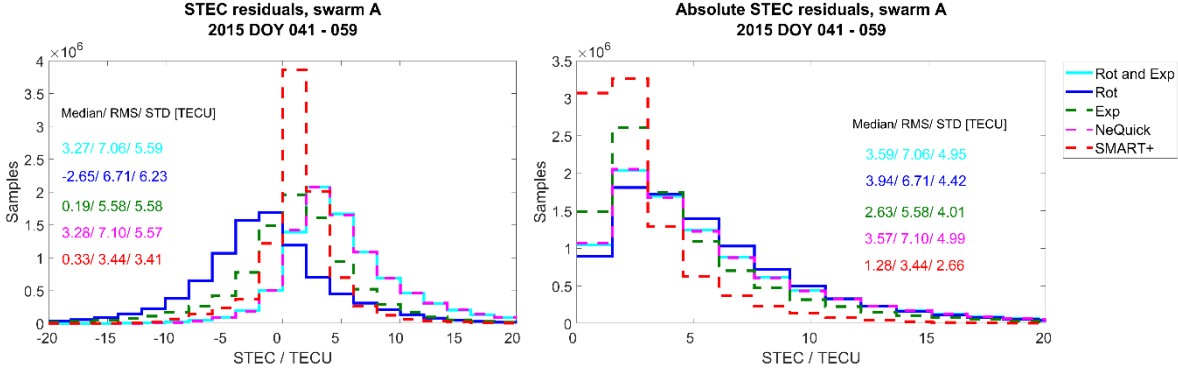


**Figure 13: Histograms of the STEC residuals (left) and absolute residuals (right) during the quiet period,**
**for Swarm A.**

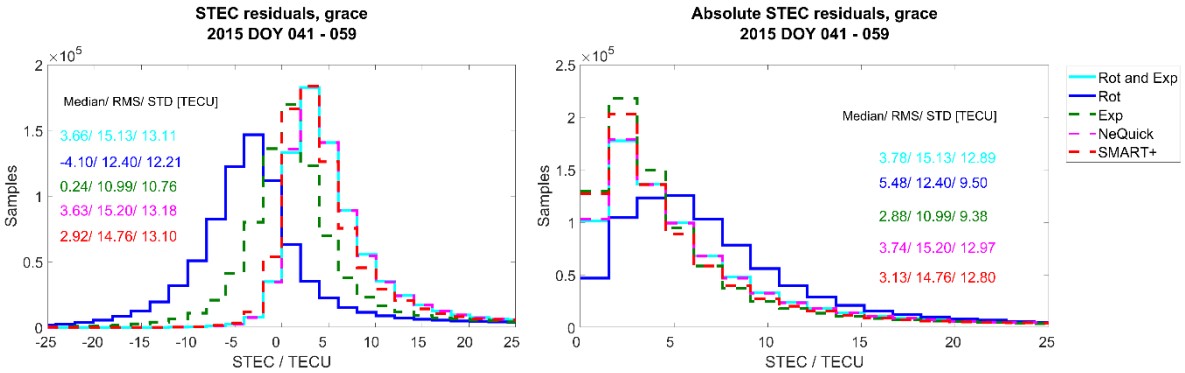

**Figure 14: Histograms of the STEC residuals (left) and absolute residuals (right) during the quiet period,**
**for GRACE.**

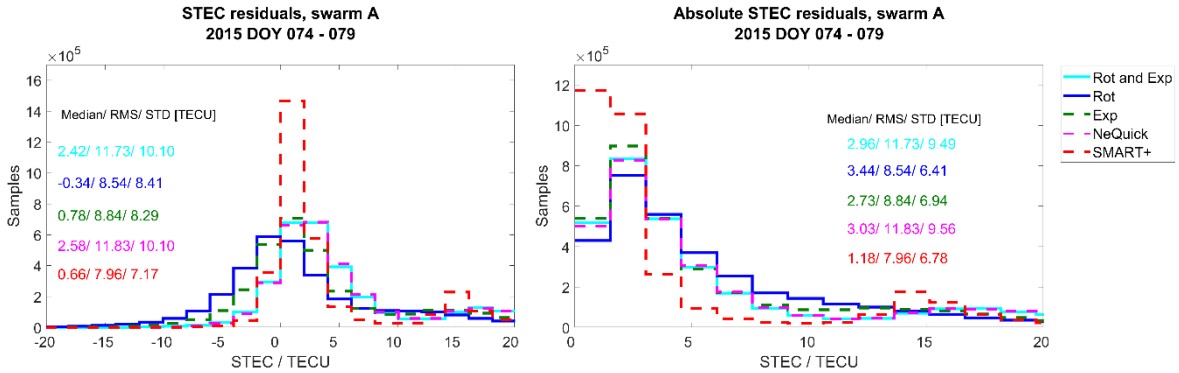

**Figure 15: Histograms of the STEC residuals (left) and absolute residuals (right) during the perturbed**
**period, for Swarm A.**

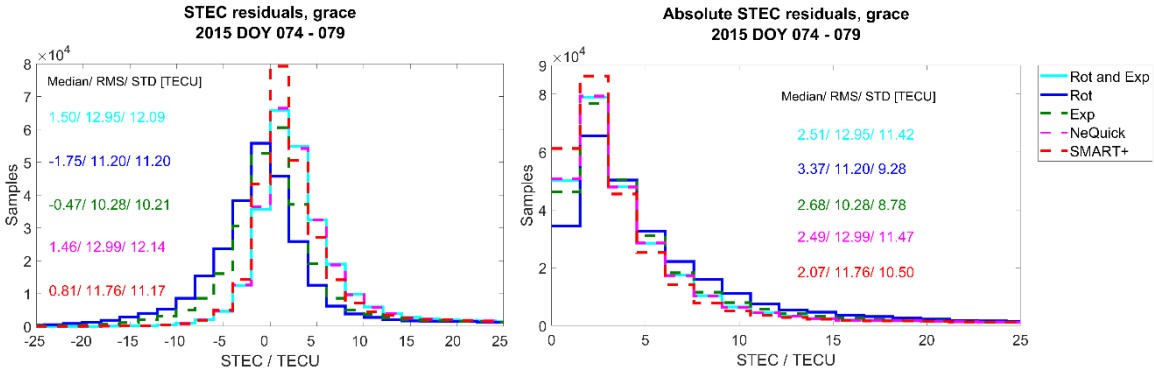

**Figure 16: Histograms of the STEC residuals (left) and absolute residuals (right) during the perturbed**
**period, for GRACE.**

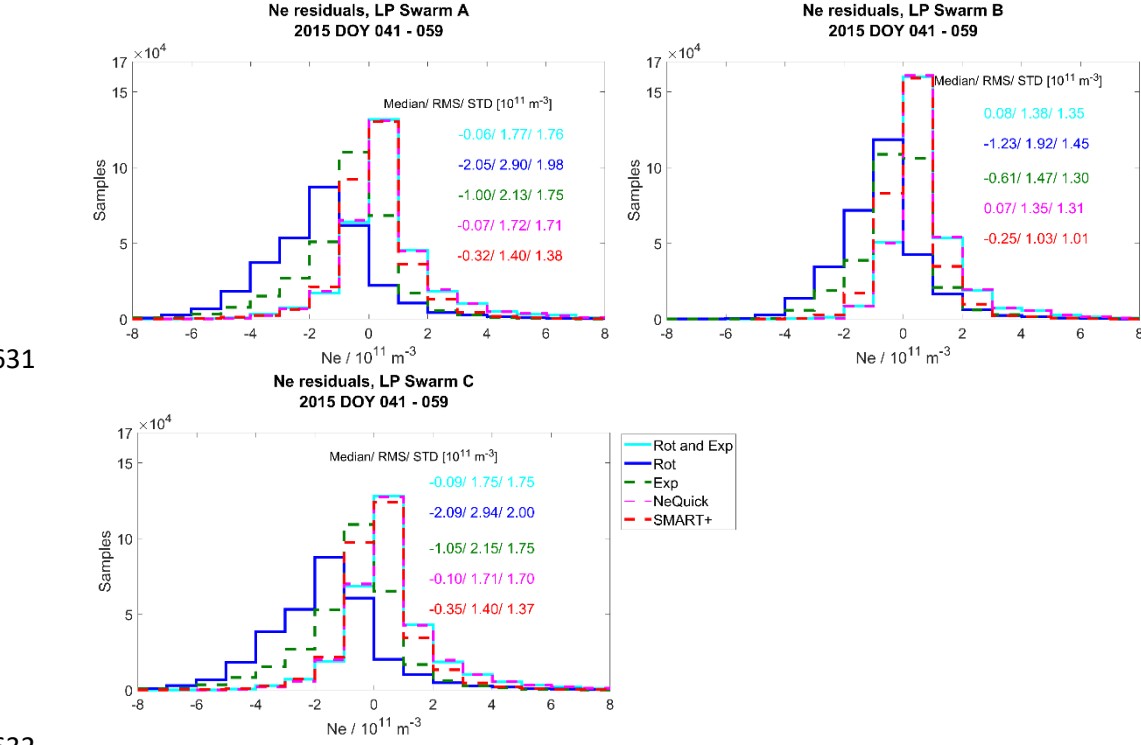



**Figure 17: Validation with LP data – distribution of the Swarm A, B, C (separated) electron density residuals for the quiet period.**

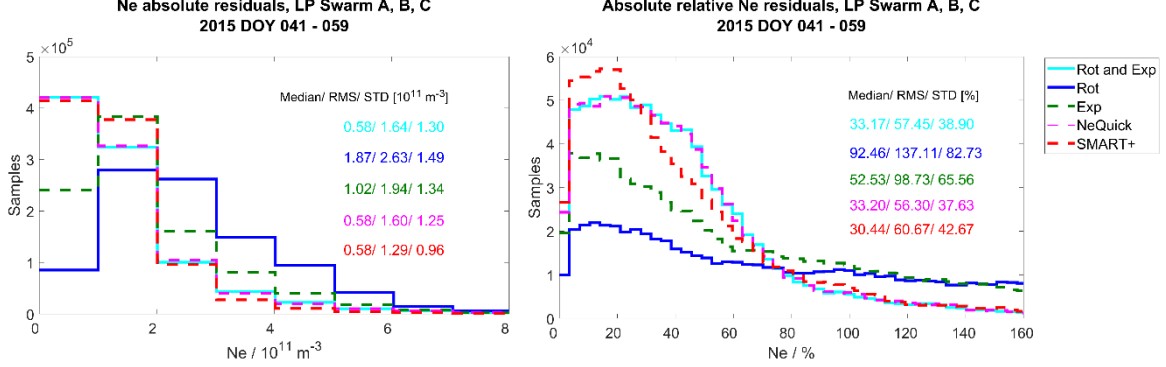


**Figure 18: Validation with LP data – distribution of the Swarm absolute and absolute relative electron density residuals for the quiet period.**

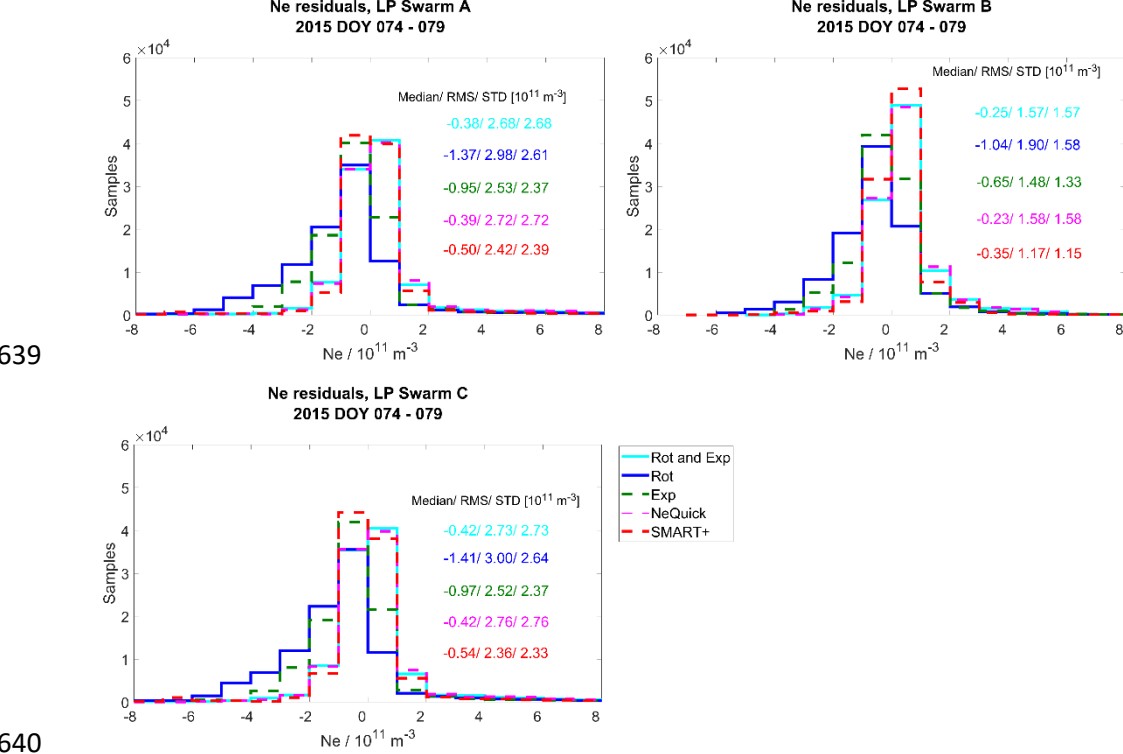

**Figure 19: Validation with LP data – distribution of the Swarm A, B, C (separated) electron density residuals for the perturbed period.**

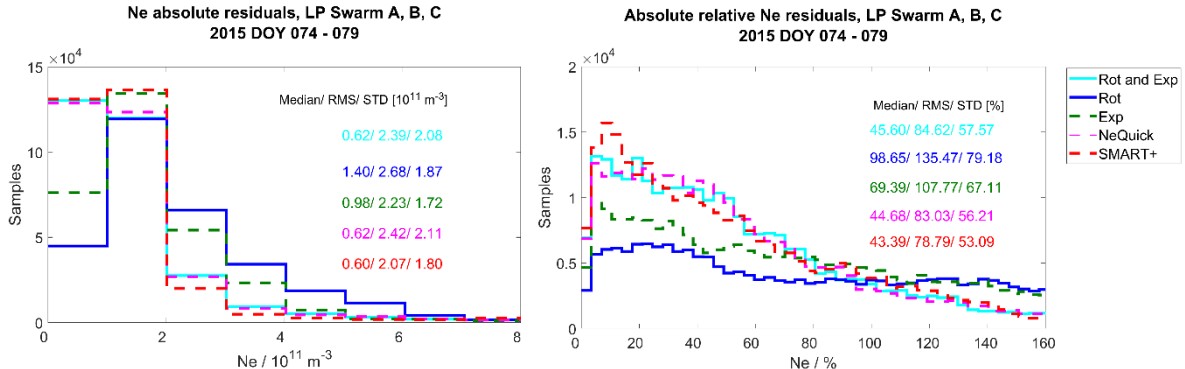

**Figure 20: Validation with LP data – distribution of the Swarm absolute and absolute relative electron density residuals for the perturbed period.**