# Peer review of "Analysis of different propagation models for the estimation of"

_Annales Geophysicae, 2020_

## Referee Comment (RC1) · Anonymous Referee #1 · 8 Jul 2020

**1) General Comments**

The manuscript "*Analyses of different propagation models for the estimation of the topside ionosphere and plasmasphere with an Ensemble Kalman Filter*" by Tatjana Gerzen et al. subjects the topic to propagate the state vector in an Ensemble Kalman Filtering (EnKF) process from one analysis step to the next, here applied to the assimilation of slant Total Electron Content (sTEC) observation data into 3D electron density grids of the topside ionosphere & plasmasphere. – Thereby investigations are focussed on statistics-based approaches, as potential alternative to more computer-intense physics-based methods to propagate the ionosphere's & plasmasphere's state from one update epoch to the next, in order to establish a new background for the next epoch's new state vector computation.

The work presented in the manuscript is set well into context with other activities concerning 3D ionospheric data assimilation techniques and modelling, that have been/are conducted in the past, more recently and actually. The presentation of algorithms and testing scheme & results appears in principle concise and clear (see also Point 2) – specific Comments). The manuscript length is adequate.

**2) Specific Comments**

Concerning the approaches to propagate the state vector by "Rotation", by "Exponential decay", or by a combination of both, it is not really clear to me, in which reference system the assimilation is conducted. – If being conducted in an earth-fixed frame, e.g. geographic, "Rotation (+ Exponential decay)" would make sense to propagate the state vector from one epoch to the next. If being conducted in a reference system being fixed w.r.t. the Sun, e.g. what the authors denote as "magnetic coordinates" (I suppose what's meant here is the Solar-Magnetic (SM) system (https://arxiv.org/ftp/arxiv/papers/1611/1611.10321.pdf, therein Section 3.5, or something similar) then "Exponential decay" would make sense. Might this be the reason why "The method Rotation delivers much higher values than NeQuick" (manuscript lines 237 – 238). – I am personally would implement a system in which the assimilation grid is maintained in SM coordinates, i.e. being fixed w.r.t. the Sun and no rotation, only exponential decay. Then, at each assimilation epoch, the observation data would for its assimilation be transformed from earth-fixed into SM, i.e. by converting the positions at which the observables were taken (in case of sTEC, position of slant range start and end point) from earth-fixed into SM. And for products generation, e.g. of TEC/Ne maps, out of this 3D SM assimilation grid, I would for requested epochs transform earth-fixed grid points, e.g. of an IONEX grid, into SM, then interpolate Ne-values/vertically integrate TEC values in the SM grid, and finally assign them to their earth-fixed positions. – So the reference frame used for the assimilation process should be clearly specified in the manuscript.

As far as I understand, the computations are done with 100 ensembles. This appears at least quite disk space-intensive to me (but probably also w.r.t. computational effort). – So it might be helpful, in this regard, to specify a bit more the computational benefits of the statistics-based approach, presented in the manuscript, over physics-based methods to propagate the state vector.

Perhaps it should be explained how the formula for *ratio*, Eq. (6), has been derived and under which criteria, e.g. the Factor 3 in its denominator.

Lines 268 – 269: STD and RMS are also higher during the quiet period, numbers listed in Fig. 4. – Do you perhaps mean "significantly higher" in relation to median, i.e. in terms of the ratios (RMS/median) and (STD/median)?

Lines 289 – 291: Where do these big sTEC differences between Swarm A (105 and 170 TECu) and GRACE (282 and 264 TECu) come from? Do you have an explanation for this?

Lines 318 – 319: I think this is not always so. Please specify more detailed.

Lines 353 – 358: See my comments above.

**3) Technical Corrections**

Line 36: "insight" should be replaced by "inside".

Line 51: The reference "Zeilhofer et al., 2010" is obviously given in the reference list as "Zeilhofer et. al., 2009" (Lines 493 – 494) – so 2010 or 2009?

Lines 56 – 57 and 69: Some braces are missing when citing references "(cf. …)" instead of "cf. …".

Eq. (8): I suppose the denominator of 20 in the square root term stands for the update rate of 20 minutes?

Eq. (9): Where does the factor 8/10 come from (which is by the way 4/5)?

Line 154: Replace "bevor" by "before".

Line 157: Instead of 0.5/100, I would write 1/200, to be more illustrative (and eventually 2/200 instead of 1/100). How are the 0.5/100 and 1/100 derived from *ratio*, i.e. from Eq. (6)?

Line 159: "… as follows: For …", i.e. capital "F".

Line 161: "… In detail …" not "… In details".

Line 162: "… = 100 x 1 and calculate to calculate …". Remove "and calculate" from this phrase.

Line 164: "… In detail …" not "… In details".

Line 209: "… Both data providers supply …", i.e. plural of "provider".

Lines 213 – 214: "… Further, information on the pre-processing of the LP data is made available." – By whom, where and when? – Probably you mean "In addition, further information on the pre-processing of the LP data is made available on this website".

Line 238: "… values seem to be …", not "seems".

Line 284: "… It is to mention here that in 2015 …", i.e. include "in".

Line 287: "For each of the tree LEOs …". "tree" should be replaced by "two" (Swarm A and GRACE) (instead of "tree" you probably meant "three").

Line 306: "~7" TECu. I think this should be rather "~8" TECu (7.96 TECu in the plot).

Line 378: "… therefore …", i.e. with an "e" at the end.

Line 506: "Figure 3: Subfigures top: …". "top" can be removed here, since Fig. 3 has no bottom row.

Plots: NeQuick is displayed in pink, SMART+ in red. These rather similar colours make it sometimes difficult to distinguish both curves. So NeQuick might be displayed in a more different colour, e.g. yellow.

\*\*\* I couldn't check all numbers (Median / RMS / STD) in the plots and their percental ratios in the text, but selected and proved only a few of them randomly.

**4) Rate:**

All in all, I would consider the scientific contribution of the manuscript as "**fairly important**".

---

## Referee Comment (RC2) · Anonymous Referee #2 · 9 Jul 2020

I think this is a very interesting, and potentially, very impactful paper.

Whilst the paper would be suitable with only minor corrections, I think if a little more analysis is done as well it would substantially increase the impact of the paper.

Please see my attached marked up copy of the document correcting English spelling and grammar and embeds all of minor comments. My major comments are:

* You need to decide how "much" you want to discuss SMART+ in this work. I understand the point of using it as a reference, but your result discussion focuses heavily on the fact that SMART+ is the best model (and you give detailed analysis of the statistics). However there is no description of the model in the paper (you have to follow

the references to find out more). If you want to keep the SMART+ discussion then I suggest you give some time to describe SMART+ and in particular the way the model time propagates - which is the main focus of this paper.

* I cannot understand how the Rot + Exp method shows almost no difference from the background NeQuick version. The Rot+Exp method is assimilating data, whereas the NeQuick results are not. The results suggest that the DA is doing nothing to the background model, and changing the time propagation method shouldn't completely negate the impact of the DA.

* In a related point, I am surprised by the amount of differences between each of the methods. I would have expected the DA to be the dominate term, with the changes caused by the propagation being minor. Since this not seem to be the case I think it needs to be carefully described why this is.

* What I would most like to see, and what I think would have the biggest impact on making this paper more cite-able in the future, is clearer analysis about the propagation results. So far the results are described in a global modelling sense - which is useful. However I am interested in understanding how each of the propagation methods work in more detail. For example you have picked 20mins propagation time but it would be interesting to see a histogram of the errors, for each technique, overplotted with increasing time, say 5min, 15min, 30min, 60min (or something similar). I would expect all methods to do very well on short time scales, but perhaps one or the other would be better at the longer scales.

You have provided some rigorous statistical results between using the different methods, but it is still hard to get a "feel" for it. I would find it interesting to see the difference between $t(n)$ and $t(n+1)$ for each method. So following the DA step (resulting in $X_a$) you could apply each method separately, and show a difference from SMART+ (as that is the best model). We would then be able to see an example of the differences (and what changes there are globally). Alternatively you could show one vertical profile
(after assimilation) and then show the impacts caused by the 3 propagation methods.

* In all cases comparing the methods to persistence (let $X\_a\hat{}t = X\_b\hat{}\{t+1\}$) would also help to demonstrate the models effectiveness. This would be especially useful if you include analysis on different timescales. For example we would expect that over 5 minutes the persistence assumption will likely work well. But at 60 mins not. It is not clear what happens on a 20min scale.

Overall I think further analysis of the results of the propagation methods, in terms of the differences found between them should be further highlighted. Time propagation for ionospheric models, without a physics-based model, is clearly of interest and so I think it is worthwhile trying to make this paper as useful as possible.

Please also note the supplement to this comment:
https://angeo.copernicus.org/preprints/angeo-2020-39/angeo-2020-39-RC2-supplement.pdf

[Figure]

**Supplement:**

[revised manuscript text omitted]

---

## Author Comment (AC1) · 21 Jul 2020

Respond to the Anonymous Referee #1

In general: Thanks a lot for your comments and suggestions!

To the "Specific comments":

1. The reference systems, we use here, are the geodetic (geographic, cf. Section 2 in
   https://arxiv.org/ftp/arxiv/papers/1611/1611.10321.pdf ) and geomagnetic (for the rotation, cf.
   Section 3.1 in https://arxiv.org/ftp/arxiv/papers/1611/1611.10321.pdf ). We will add this
   information to the manuscript.
2. You are right, the storage of 100 ensembles is indeed quite disk space-intensive. One analyzed
   solution with 100 ensembles needs around 150 MB. However, I think, it is not necessary to store
   all the ensembles of the analyzed solutions over the whole period of time, one is interested in.
   Ones we propagated an analyzed ensemble matrix in time, one could calculate the ensemble
   mean (which represents the reconstructed Ne for this time stamp) and delete the ensemble
   matrix. Regarding the computational effort, it is not high, and it depends a lot on the
   propagation method. Because the calculation of the ensembles by the empirical background
   model is a bit time-consuming and depends on the model, one uses. In general, to apply a
   relatively easy manageable empirical model, like NeQuick (which needs just F10.7 index as
   external input), is much easier, than to run a physical model, like TIEGCM
   https://ccmc.gsfc.nasa.gov/models/modelinfo.php?model=TIE-GCM , with all the inputs one
   need. Our Analysis Step (data assimilation) plus the Forecast Step cost together around 1 minute
   on a Linux machine.
3. The formula for ratio in Eq. 6 stands for relative error caused by using of $Rot\left(x^b(t_n)\right)$ instead of
   $x^b(t_{n+1})$ and represents in this way the relative mean error introduced by approximation of the
   true state at time $t_{n+1}$ by a simple rotation of the true state at time $t_n$. Some of the factors in
   the formulas, like 3 in Eq. 6; 8/10 in Eq. 9 were derived by try-and-error method. This means, we
   did several runs with different factors and then validate the results. At the end, the factors
   delivering the best results were chosen for the paper. We will put a corresponding clarifying note
   in the paper.
4. Lines 268 – 269: it is indeed written a bit confusing in the manuscript. We will correct it.
5. Lines 289 – 291: It is indeed a big difference. The only explanation we see is that the GRACE
   satellites were flying below the SWARM A, and thus near on the hmF2 value. We checked again
   the orbit heights of SWARM A and GRACE satellites for the periods of interest within 2015. We
   found a small but in this context probably important mistake - the GRACE orbit was around 430
   km (not 450 km, as was given by us in the paper). We correct the value in the paper.
6. Lines 318 – 319: you right, thanks. This is only true for the SWARM A residuals. We add the
   correction to the manuscript text.
7. Eq. (8): exact.
8. Eq. (9): please see number 3. above.
9. Line 157: we wrote 0.5/100 and 1/100 to make more clear that we mean one-half and one
   percent. The factors are chosen by try-and-error.
10. Line 306: The STD value in the Fig. 11 is 7.17 TECU, this is why we round it to 7 TECU.

---

## Author Comment (AC2) · 26 Aug 2020

Respond to the Anonymous Referee #2

In general: Thanks a lot for your comments and suggestions and especially for your helpful major comments!

To the major comments:

1. *You need to decide how "much" you want to discuss SMART+ in this work.:* We have added a subsection with the most important information about SMART+.

2. *I cannot understand how the Rot + Exp method shows almost no difference from the background NeQuick version.:*
   We tuned all the tested models with respect to their error models (process noise, systematic error, weights between X_a and the background model) and did several runs (up to 15 not show in the paper) to figure out the best configurations. When we increased the error terms, the results became unreasonable after some days of EnFK.  At the end, the presented variants of the methods delivered the best results (reasonable and providing the smallest validation residuals). However, we think that especially for the Rot+Exp method the manuscript-presented choice of the error terms and weighting parameters could be still not optimal. For this method, these errors are probably too small, and that is the reason why the differences from the background model are small. We think about an adaptive method to calculate these errors, but this will be future work which cannot be covered in this paper.

3. *In a related point, I am surprised by the amount of differences between each of the methods. I would have expected the DA to be the dominate term, with the changes caused by the propagation being minor. Since this not seem to be the case I think it needs to be carefully described why this is.*
   We think that the main reasons therefore are the following: (1) Compared with the huge topside iono & plasmasphere volume, there are not much measurements available (compared for example to the situation when you apply data assimilation using all the available ground based data). (2) Using sequential data assimilation one has to deal with changes summed up with time. At the beginning of the period you are interested in (for example in our case, on DOY 041, early morning hours), the differences between the results using different propagation methods are not so big, but then they become more, because the propagation goes in each time step complete different directions. Like e.g. using Rotation you just rotate the analysed solution, but using Rotation with exponential decay you build in each propagation step a weighted sum between rotated analysed solution and background.

4. *For example you have picked 20mins propagation time but it would be interesting to see a histogram of the errors, for each technique, overplotted with increasing time, say 5min, 15min, 30min, 60min (or something similar).*

   We investigated the EnKF as well as SMART+ regarding their ability to reproduce assimilated STEC as well as to estimate independent STEC measurements and in-situ electron density measurements in dependency on changed temporal resolution of 60 minutes. For all tested methods as well as for NeQuick model we obtained no significant differences in the statistics between the validation results for $\Delta t=60$ versus $\Delta t=20$ minutes. We think, this indicates that the presented propagation methods work in generally well. Contrary, for example for simple persistence as propagation method, the reconstruction results become implausible more quickly if $\Delta t$ is enlarged.

5. *I would find it interesting to see the difference between t(n) and t(n+1) for each method. So following the DA step (resulting in X_a) you could apply each method separately, and show a difference from SMART+ (as that is the best model). We would then be able to see an example of the differences (and what changes there are globally).*

We have added Figure 7 showing for all the EnKF variants the differences $E\left(X^f(t_{n+1})\right) - E\left(X^a(t_n)\right)$ on the left and the percentage differences on the right calculated as $100 \cdot$

$$\cdot \left(E\left(X^f(t_{n+1})\right) - E\left(X^a(t_n)\right)\right) \Big/ \left[E\left(X^f(t_{n+1})\right) + E\left(X^a(t_n)\right)\right] \text{ for a fixed } t_n.$$

6. *In all cases comparing the methods to persistence (let X_aˆt = X_bˆ{t+1}) would also help to demonstrate the models effectiveness.*

We indeed tried also "persistence" as a propagation method. But "persistence" delivered very bad results. Already after ca. 24 hours, reconstructions based on "persistence" had shown unreasonable effects, like completely misplaced equatorial crest region. The propagation time of 20 minutes seems to be too long for the method "persistence". We have added this information to the paper.

To the technical comments:

1. We put two Ref. to the statement "Around 50% of the signal delays…" Line 28.

2. Regarding your questions on the factors which we had chosen within the equations of the forecast methods, like 8/10 in Eq. (9): We had derived them by an internal validation. Particularly, we had conducted several runs with different factors and then analyzed the results. At the end, the factors delivering the best results have been applied in the paper. We have added a corresponding note in the paper.

3. Regarding your question about the generation of the ensembles, Line 170ff: We chose 3/100 based on our analysis of the sensitivity of the NeQuick model on F10.7 (we tested also higher factors). We did this analysis indeed only around the periods important for the paper. Thus, the factor 3/100 could be a different for others periods. We think that the generation of the ensembles with respect to their variation and amount forms another important topic to apply the EnKF (in addition to the choice of the propagation method). Until now, we have not found publications in the field of ionospheric research focusing on this topic. We would be very thankful for a reference.

An extensive validation of the reconstruction results in dependence on the number of ensembles and the method to generate them is in work but not covered in the scope of this paper.

4. Fig. 2, top subfigures, (your question at line 228) shows the reconstructed results for DOY 076, 19:00 UT. In order to derive these figures: (1) We take the analysed ensembles from UT 18:40 and (2) propagate those 20 minutes in time using "Rot and Exp". (3) We update the propagated ensembles by the assimilation of the currently available (around 19:00 UT) STEC measurements (analysis step of EnKF). (4) We calculate the ensemble mean, which is partly shown for a fixed latitude/longitude/height in Fig. 2. The described propagation-update-cycle, which has been used to obtain the results shown in this figure, had started at time 0:20 UT on DOY 041 and has been repeated until 19:00. Consequently, data has been assimilated and "Rot+Exp" has been applied.

5. Your comments to lines 234, 238: We have added figures 4, 5, showing the differences and the percentage differences between the reconstructed and the modeled electron densities; and figure 6, illustrating the corresponding measurements geometry.